# The proportion of randomized controlled trials that inform clinical practice

**Nora Hutchinson[1], Hannah Moyer[1], Deborah A Zarin[2], Jonathan Kimmelman[1]\***

[1]Studies of Translation, Ethics and Medicine (STREAM), Dept. of Equity, Ethics and Policy, McGill University, Montreal, Canada; [2]Multi-Regional Clinical Trials Center of Brigham and Women's Hospital and Harvard, Boston, United States

**Abstract** Prior studies suggest that clinical trials are often hampered by problems in design, conduct, and reporting that limit their uptake in clinical practice. We have described 'informativeness' as the ability of a trial to guide clinical, policy, or research decisions. Little is known about the proportion of initiated trials that inform clinical practice. We created a cohort of randomized interventional clinical trials in three disease areas (ischemic heart disease, diabetes mellitus, and lung cancer) that were initiated between January 1, 2009 and December 31, 2010 using ClinicalTrials.gov. We restricted inclusion to trials aimed at answering a clinical question related to the treatment or prevention of disease. Our primary outcome was the proportion of clinical trials fulfilling four conditions of informativeness: importance of the clinical question, trial design, feasibility, and reporting of results. Our study included 125 clinical trials. The proportion meeting four conditions for informativeness was 26.4% (95% CI 18.9–35.0). Sixty-seven percent of participants were enrolled in informative trials. The proportion of informative trials did not differ significantly between our three disease areas. Our results suggest that the majority of randomized interventional trials designed to guide clinical practice possess features that may compromise their ability to do so. This highlights opportunities to improve the scientific vetting of clinical research.

## Editor's evaluation

This article constructs a rigorous four-step assessment of the informativeness of a clinical trial that measures its feasibility, reporting, importance, and risk of bias. This work is highly relevant for the class of trials for which it is defined, namely clinically directed randomized controlled trials. It could also be translated and validated in other areas, using data from a wider set of sources beyond the trial registry clinicaltrials.gov. The extended longitudinal nature of the assessment and the potential for some subjectivity limit this tool's utility to being a retrospective 'thermometer' for measuring informativeness rather than as a prospective diagnostic and/or fix for at-risk designs.

**\*For correspondence:**
jonathan.kimmelman@mcgill.ca

## Introduction

The ultimate goal of clinical research is to produce evidence that supports clinical and policy decisions. Numerous analyses suggest that a substantial proportion of clinical trials aimed at informing clinical practice are marred by flaws in design, execution, analysis, and reporting (*Chalmers et al., 2014*; *Chan et al., 2014*; *Fergusson et al., 2005*; *Glasziou et al., 2014*; *Ioannidis et al., 2014*; *Al-Shahi Salman et al., 2014*; *Yordanov et al., 2018*; *Yordanov et al., 2015*). The initial research response to COVID-19 illustrated the fact that existing oversight mechanisms fail to prevent the initiation of flawed trials (*Bugin and Woodcock, 2021*). While unexpected events can stymie well-conceived and implemented studies, trials that have features rendering them unlikely to inform clinical practice may do

harm by misleading potential participants about their benefits and by diverting patient-participants from otherwise informative research efforts (*Zarin et al., 2019*).

We have previously described five conditions that trials should fulfill to support clinical or policy decision-making (*Zarin et al., 2019*; *London and Kimmelman, 2020*). First, trials must ask an important and clinically relevant question that is not yet resolved. Second, trials must be designed to provide a meaningful answer to that question. Third, trials must be feasible, with achievable enrollment goals and timely primary outcome completion. Fourth, outcomes must be analyzed in ways that support valid interpretation. Last, trial results must be made accessible in a timely fashion.

In what follows, we created surrogate measures for four conditions of informativeness: trial importance, design quality, feasibility, and reporting (the fifth condition, analytical integrity, did not lend itself to objective, dichotomous assessment, and is not assessed below). We then evaluated the proportion of clinically directed randomized trials in three common disease areas meeting these four conditions. Our approach involved a retrospective evaluation of trial informativeness. Findings

**Table 1.** Characteristics of intervention trial cohort.

| Category | Ischemic heart disease trials N=40 | Diabetes mellitus trials N=57 | Lung cancer trials N=28 | All trials N=125 (%) |
|---|---|---|---|---|
| Trial phase | | | | |
| 2* | 6 (15.0) | 5 (8.8) | 13 (46.4) | 24 (19.2) |
| 3[†] | 11 (27.5) | 26 (45.6) | 13 (46.4) | 50 (40.0) |
| 4 | 10 (25.0) | 9 (15.8) | 0 (0.0) | 19 (15.2) |
| NA[‡] | 13 (32.5) | 17 (29.8) | 2 (7.1) | 32 (25.6) |
| Intervention | | | | |
| Drug/biologic | 19 (47.5) | 34 (59.6) | 24 (85.7) | 77 (61.6) |
| Combination[§] | 7 (17.5) | 0 (0.0) | 1 (3.6) | 8 (6.4) |
| Device | 4 (10.0) | 4 (7.0) | 0 (0.0) | 8 (6.4) |
| Other[¶] | 10 (25.0) | 19 (33.3) | 3 (10.7) | 32 (25.6) |
| Trial status | | | | |
| Completed | 29 (72.5) | 53 (93.0) | 17 (60.7) | 99 (79.2) |
| Terminated | 7 (17.5) | 1 (1.8) | 7 (25.0) | 15 (12.0) |
| Active, NR | 0 (0.0) | 1 (1.8) | 4 (14.3) | 5 (4.0) |
| Unknown | 4 (10.0) | 2 (3.5) | 0 (0.0) | 6 (4.8) |
| Outcome | | | | |
| Clinical | 24 (60.0) | 8 (14.0) | 10 (35.7) | 42 (33.6) |
| Surrogate | 16 (40.0) | 49 (86.0) | 18 (64.3) | 83 (66.4) |
| Sponsor** | | | | |
| Industry | 18 (45.0) | 27 (47.4) | 13 (46.4) | 58 (46.4) |
| Other[††] | 22 (55.0) | 30 (52.6) | 15 (53.6) | 67 (53.6) |

*Including phase 1/2.

[†]Including phase 2/3.

[‡]Includes behavioral, procedural/surgical, and device interventions .

[§]Including Drug + Device, Drug + Procedure, Behavioral + Device, Radiation Therapy + Drug.

[¶]Including Behavioral Intervention, Radiation Therapy, Surgical Procedure, Cellular Intervention.

**As defined in ClinicalTrials.gov registration records.

[††]Included within the designation 'Other' are seven trials that received funding from the U.S. National Institutes of Health (NIH) or other U.S. Federal agencies, and 60 trials that are non-industry and non-NIH/U.S. Federal agency funded.

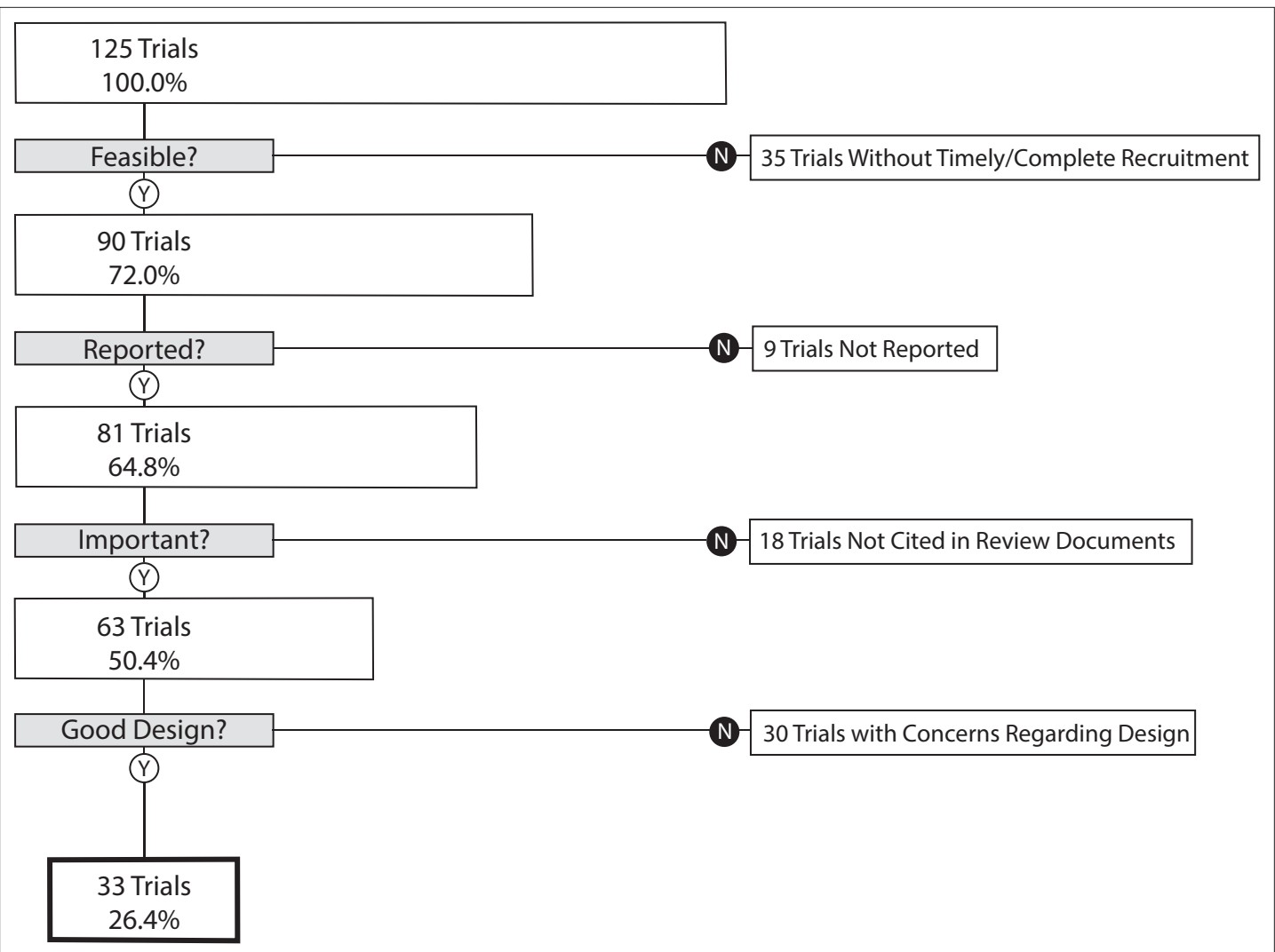

**Figure 1.** Flow diagram – the proportion of trials meeting four conditions of informativeness.

can help healthcare and research systems identify study types in need of further scrutiny, thereby improving the impact of future research.

## Results

Over half of the 125 interventional trials in our cohort were studies of drug or biologic interventions (77 trials; 61.6%). The majority were phase 2 (24 trials, 19.2%) or phase 3 trials (50 trials, 40.0%). Trial status was 'completed' in 99 of 125 trials (79.2%) and 'terminated' in 15 trials (12.0%) (*Table 1*). Ninety-three trials (74.4%) were first registered on ClinicalTrials.gov prior to or within 30 days of the listed trial start date.

Our primary outcome, the proportion of trials that informed clinical practice, was 26.4% (95% CI 18.9–35.0) (*Figure 1*). As a sensitivity analysis, we re-analyzed our primary outcome excluding the 35 trials in the lowest quartile for target enrollment. This resulted in a proportion of informative trials of 35.6% (95% CI 25.7–46.3). We performed a second sensitivity analysis on our primary outcome excluding phase 1/2 and phase 2 trials. This resulted in 30.7% (95% CI 21.9–40.7) of trials meeting four conditions of informativeness.

A total of 193,839 participants were enrolled in the 125 trials in our cohort, of which 129,973 (67.1%) were enrolled in informative trials. The proportion of ischemic heart disease trials that was informative was 27.5% (95% CI 14.6–43.9); the proportion for diabetes mellitus trials was 31.6% (95%

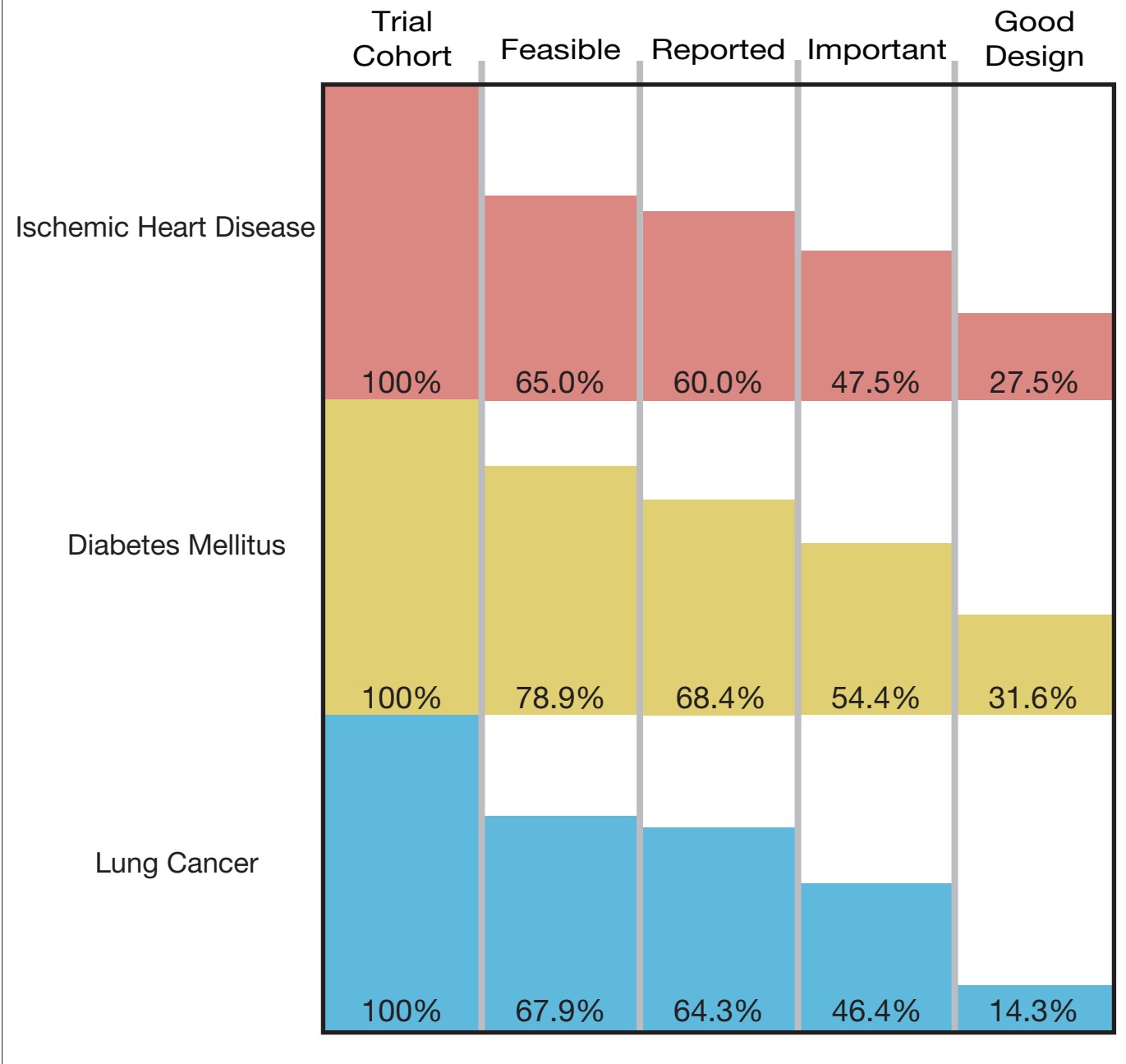

**Figure 2.** The cumulative proportion of trials meeting four conditions of informativeness by disease area.

The online version of this article includes the following figure supplement(s) for figure 2:

**Figure supplement 1.** The cumulative proportion of trials meeting four conditions of informativeness by sponsor.

CI 19.9–45.2), and the proportion for lung cancer was 14.3% (95% CI 4.0–32.7) (*Figure 2*). Proportions did not vary significantly by disease area (p-value = 0.23) (*Table 2*). Each surrogate measure contributed considerably to the stepwise decline in the proportion of informative trials (*Supplementary file 2*).

Of the 35 trials not meeting our feasibility condition, 21 failed to reach 85% of planned patient enrollment (*Supplementary file 3*). Of the nine trials not meeting our reporting condition, six were not subject to FDAAA 801 (*U.S. Public Law, 2007a*) results reporting requirements (*Supplementary file 4*). Although we cannot be certain why a trial was not incorporated into a clinical synthesizing document, possible reasons are illustrated by the following examples. For one trial (NCT01104155 – phase

**Table 2.** The proportion of informative trials by trial property.

| Category | Proportion of informative trials | 95% CI | p Value |
|---|---|---|---|
| Trial phase | | | |
| 2* | 8.3 | 1.0–27.0 | |
| 3† | 50.0 | 35.5–64.5 | |
| 4 | 10.5 | 1.3–33.1 | |
| NA | 12.5 | 3.5–29.0 | $5.2 \times 10^{-5}$ |
| Intervention | | | |
| Drug/biologic | 35.1 | 24.5–46.8 | |
| Combination‡ | 25.0 | 3.2–65.1 | |
| Device | 0.0 | 0.0–36.9 | |
| Other§ | 12.5 | 3.5–29.0 | $2.0 \times 10^{-2}$ |
| Disease area | | | |
| Ischemic heart disease | 27.5 | 14.6–43.9 | |
| Diabetes mellitus | 31.6 | 19.9–45.2 | |
| Lung cancer | 14.3 | 4.0–32.7 | 0.2 |
| Sponsor¶ | | | |
| Industry | 50.0 | 36.6–63.4 | |
| Non-industry | 6.0 | 1.7–14.6 | $8.1 \times 10^{-8}$ |

*Including phase 1/2.
†Including phase 2/3.
‡Including Drug + Device, Drug + Procedure, Behavioral + Device, Radiation Therapy + Drug.
§Including Behavioral Intervention, Radiation Therapy, Surgical Procedure, Cellular Intervention.
¶As defined in ClinicalTrials.gov registration records.

2 study in non-small cell lung cancer investigating the combination of eribulin mesylate in combination with intermittent erlotinib), increased understanding of the importance of biomarker status in treatment response from trial initiation to completion rendered the results for a non-selected population less clinically relevant (*Supplementary file 5*; *Mok et al., 2014*). In another case, trial NCT00918138 investigated the addition of saxagliptin to extended-release metformin in adult patients with type 2 diabetes (*Neutel et al., 2013*). Its primary outcome, change from baseline in 24-hr mean weighted glucose at week 4, did not meet the eligibility criteria for inclusion in a high-quality systematic review (SR) investigating the same topic (*Men et al., 2018*). In a third case, trial NCT00954707 (a phase 4 trial investigating duration of dual anti-platelet therapy in 2509 individuals undergoing placement of a Cypher cardiac drug eluting stent) sponsors submitted results to ClinicalTrials.gov but never published them. This may have hampered inclusion in a clinical synthesizing document. Out of the 18 trials, 8 trials not meeting the importance condition had no published primary outcome results. Finally, the most common reasons for a high risk of bias (ROB) score in the 30 trials not meeting our design condition were lack of blinding of participants and personnel, and lack of blinding of outcome assessment and selective reporting (*Supplementary file 6*).

Studies sponsored by industry were significantly more likely to fulfill all four conditions of informativeness than those not sponsored by industry (50.0 vs. 6.0%, p-value <0.001) (*Table 2*; *Figure 2— figure supplement 1*). Using the two-sided Fisher's exact test, there was a non-random association between trial phase and informativeness, and type of intervention and informativeness (*Table 2*).

## Discussion

This study provides the first assessment of the proportion of randomized trials fulfilling four key conditions of informativeness. In our analysis, just over one-fourth of trials demonstrated adequacy for study feasibility, reporting, importance, and design. The remaining 73.6% contained a limitation in design, conduct, or reporting that compromised their ability to inform clinical decision-making.

Certain shortcomings of clinical trials are a result of experimenting in a dynamic real-world environment and cannot be entirely avoided. Clinical trials are difficult to plan, and there may be defensible reasons for falling short of some conditions. For example, changes in medical practice may render a research question irrelevant to clinical practice; an emerging pandemic might lead to underrecruitment. However, our findings underscore the major challenges sponsors and clinical investigators confront in fulfilling the scientific and ethical warrant for enrolling patient-participants in randomized trials. The goal should be to address foreseeable limitations in trial design, conduct, or reporting. For example, increased oversight by research funders, including requirements for landscape analysis of completed and ongoing clinical trials to ensure trials are addressing important questions, and the provision of independent scientific review to highlight vulnerabilities in trial design (*Bierer et al., 2018*), are measures that can be implemented to increase the likelihood that trials will be informative. Many methodological weaknesses in trial design can be corrected at minor cost (*Yordanov et al., 2015*).

The proportion of informative trials did not differ significantly between ischemic heart disease, diabetes mellitus, and lung cancer, indicating shared challenges in design, implementation, and reporting. Our study also demonstrated that each condition of informativeness goes unfulfilled in roughly equal proportions (*Supplementary file 2*), suggesting that vigilance is required throughout the life cycle of a trial. Our estimates for the fraction of studies fulfilling criteria for recruitment feasibility are in line with prior studies (*Carlisle et al., 2015*; *Cheng et al., 2011*; *Korn et al., 2010*; *Walters et al., 2017*). The fraction of trials at low ROB is similar to prior estimates (*Yordanov et al., 2015*; *Ndounga Diakou et al., 2017*; *Vale et al., 2013*). Our estimate for the fraction of studies fulfilling reporting requirements (90.0%) is in line with prior studies that evaluated both ClinicalTrials.gov results deposition and publication (*Chen et al., 2016*; *Phillips et al., 2017*), both of which were deemed acceptable means of results reporting in our study. To our knowledge, our study is the first to apply these conditions jointly to a sample of trials, in addition to assessing importance via citation in clinical synthesizing documents.

Our results also indicate that certain types of trials may be at greater risk for having their informativeness compromised. Although there was no significant difference in informativeness observed between our three disease areas, many lung cancer trials were of early phase (46.4% phase 2) and did not meet design criteria due to lack of blinding (*Supplementary file 6*). We acknowledge that for some of these trials, blinding may have been difficult to achieve but nonetheless would contribute to an elevated ROB. Of note, trials were assessed as 'low ROB' if there was no or incomplete blinding, but the outcome was unlikely to be influenced by lack of blinding, as per the Cochrane ROB guidelines (*Higgins et al., 2011*).

Phase 4 trials fared worse than phase 3 trials, with only 2 of 19 fulfilling all four conditions of informativeness (*Supplementary file 7*). Trials sponsored by industry funders were far more likely to fulfill all four conditions than those with non-industry sponsorship (50.0 vs. 6.0%; *Figure 2—figure supplement 1*). This is in keeping with prior research demonstrating greater recruitment challenges for non-industry funded trials (*Carlisle et al., 2015*), in addition to diminished compliance with timely results reporting on ClinicalTrials.gov (*DeVito et al., 2020*).

These results suggest that funding bodies and academic medical centers may not provide adequate resources for fulfilling the clinical mission of the trials they support. Several recent initiatives aim at improving various aspects of informativeness, including increased consideration given to the importance and clinical relevance of the research question, the evidentiary basis for proposed research, study registration and reporting, by many funders (*Moher et al., 2016*). The implementation of new frameworks, such as INQUIRE, developed to guide academic institutions in addressing waste in research, including assessments of research design, feasibility, transparency, relevance, and internal and external validity, if widely adopted, may lead to further improvements in research quality (*von Niederhäusern et al., 2018*). The SARS-CoV-2 pandemic has highlighted both the susceptibility of our clinical research enterprise to substandard trials, while also showing

what is possible with robust research vetting, coordination, and collaboration (*Kimmel et al., 2020*).

Our study should be interpreted considering several limitations. First, our measures for each condition of informativeness are proxies for the concepts they represent. For example, scoring trial importance required citation in a clinical synthesizing document. This measure may have erroneously classified some informative trials as at risk of being uninformative (e.g. trials that evaluate disease management in niche populations that are not addressed in practice guidelines or SRs). It may also have misclassified some trials as informative (e.g. trials addressing already resolved clinical hypotheses, which might nevertheless be cited in SRs). To the former, none of the 18 trials not fulfilling the importance condition involved niche populations (*Supplementary file 5*). We also acknowledge that some trials may inform clinical practice despite failing our criteria. The DAPT Study (NCT00977938) was a large phase 4 study that was deemed at high ROB in several high-quality SRs (*Xu et al., 2021*; *Yin et al., 2019*). However, this study has had an important impact on the clinical management of antiplatelet therapy following drug-eluting stent placement (*Cutlip and Nicolau, 2020*). Our metrics are best understood as capturing factors that seriously (but not necessarily fatally) compromise a trial's prospects of informing practice and that are rectifiable.

Second, we applied strict inclusion/exclusion criteria when identifying our cohort of clinically directed randomized controlled trials, thus limiting generalizability to other types of trials, including those involving diagnostics, early phase trials, nonrandomized trials or interventions that do not advance to FDA approval. The latter would require different criteria, given their primary goal of informing regulatory or research decision-making. Developing surrogate measures of informativeness for other types of trials represents an avenue for future research.

Third, we used a longitudinal and sequential approach, since some of the conditions were only relevant once others had been met. For example, incorporation into a clinical synthesizing document can only occur once results have been reported. Our sequential approach enabled us to address our primary outcome with an economy of resources. However, our study does not enable an assessment of the proportion of trials fulfilling three of the four criteria in isolation from each other. In addition, changes in research practices or policy occurring over the last decade might produce different estimates for the proportion of randomized trials that are informative.

Fourth, our evaluation is limited by the accuracy of information contained in the ClinicalTrials.gov registration record and in the published literature. The use of other sources of data, such as U.S. FDA regulatory documents, in addition to study protocols and statistical analysis plans uploaded onto ClinicalTrials.gov, could be considered for use in future evaluations of trial informativeness.

Although there is broad agreement that uninformative trials exist, there is no clear consensus on methods to identify which trials are uninformative. Our retrospective assessment is an initial step toward developing prospective criteria that can be used to highlight trials of concern, thereby enabling early corrective intervention.

Trial volunteers are generally told that their participation will advance clinical practice. However, one-third (33%) of patient-participants in our study were enrolled in trials that possessed at least one feature that compromised their goal of informing clinical practice. Sponsors and investigators often face unforeseeable challenges, and trials with flaws in design and implementation occasionally uncover actionable insights. Nevertheless, research systems and oversight should address persistent barriers to fulfilling the societal mission of clinical research.

## Materials and methods
### Overview of approach
We created a cohort of randomized, interventional clinical trials in three broad disease areas that are representative of the clinical research enterprise and that have a significant impact on patient morbidity and mortality: ischemic heart disease, diabetes mellitus, and lung cancer. We restricted inclusion to trials that appeared to be aimed at informing clinical practice by selecting trials with a stated purpose of treatment or prevention of disease and with a primary clinical outcome or appropriate surrogate. We established milestones that could serve as objectively verifiable surrogates for four conditions of informativeness. Trials in our sample were then tracked forward to assess the

proportion attaining each informativeness condition. 'Informative trials' were trials that fulfilled all four conditions of informativeness.

## Surrogate measures for four conditions of informativeness

We formulated surrogate measures for each condition of informativeness. Measures were chosen based on (i) close correspondence with each informativeness condition; (ii) objective and reproducible dichotomous scoring; and (iii) feasibility of assessment. The four surrogates of informativeness, described in greater detail below, were as follows: trial importance (determined by citation of reported trial results in high-quality clinical synthesizing documents; the premise of this surrogate is that these documents focus on questions of clinical importance); trial design quality (assessed using a modified Cochrane ROB tool, which is designed to identify threats to study internal validity); trial feasibility (established based on ability to achieve adequate participant enrollment and timely primary outcome completion); and reporting (based on accessibility of primary outcome results via deposition on ClinicalTrials.gov or in journal publications).

## Clinical trial sampling

We identified all trials registered on ClinicalTrials.gov in our three disease areas with a start date from January 1, 2009 to December 31, 2010 inclusive (*Supplementary file 8*). Our time range provided a minimum of 9 years of follow-up for maturation toward trial completion and fulfillment of all four surrogates of informativeness. Trials were downloaded from ClinicalTrials.gov on May 15, 2020. We updated trial status and enrollment for all trials meeting our inclusion criteria on October 6, 2021.

We included randomized trials (i) evaluating interventions of any type; (ii) aimed at the treatment or prevention of ischemic heart disease, diabetes mellitus, or lung cancer; (iii) with at least one site in the United States (most of which will thus have a regulatory requirement for results reporting) (*U.S. Public Law, 2007b*); and (iv) interventions that were FDA approved, that advanced to FDA approval, or interventions not subject to FDA approval (e.g. cardiac rehabilitation). We did not include trials that we deemed unlikely to be targeted at informing clinical practice by excluding: (i) studies that exclusively evaluated safety, diagnostic, or screening interventions and (ii) early phase trials (phase 0 or phase 1) (*Supplementary file 9*; *Figure 3*; *Figure 3—figure supplement 1*; *Figure 3—figure supplement 2*; *Figure 3—figure supplement 3*; *Supplementary file 10*). Phase 2 trials were included in our study as they are frequently used to inform both clinical and regulatory decision-making, particularly in cancer, where over one-quarter of recent FDA cancer drug approvals were based on the results of phase 1/2 or phase 2 clinical trials (*Tibau et al., 2018*). Trials were independently screened and assessed for eligibility by two authors (NH and HM), with disagreements resolved by a third reviewer (JK).

## Scoring conditions of informativeness

Two authors (NH and HM) independently scored all trials for the surrogate measures of the four conditions of informativeness (*Supplementary file 11*). Disagreements were resolved by a third reviewer (JK). Because of their logical relationship among surrogate measures (e.g. citation in a high-quality clinical synthesizing documents cannot be assessed unless trial results are available) and workflow (e.g. ROB information is often available in SRs), conditions were scored sequentially. Trials not meeting one condition were not advanced for evaluation of subsequent conditions. The order of scoring was as follows: (i) feasibility; (ii) reporting; (iii) importance; and (iv) design. Trials meeting all four conditions were deemed informative; trials failing on any condition possessed features that compromised their informativeness.

Our assessments of informativeness began by evaluating feasibility based on timely trial completion and patient-participant recruitment success. Completed trials were deemed to have not fulfilled feasibility if final participant enrollment was less than 85% of expected enrollment as listed in the final registration record prior to study start, thus reflecting a substantial loss of statistical power for the primary outcome (*Carlisle et al., 2015*). Terminated trials were deemed infeasible if the reason for termination in the ClinicalTrials.gov registration record involved accrual, feasibility, funding, or another non-scientific reason (*Supplementary file 12*). Trials terminated for scientific reasons (accumulated scientific data suggesting early efficacy, futility, or toxicity) were deemed feasible irrespective of the proportion of expected enrollment achieved. Trials that were ongoing were categorized as infeasible if they had already surpassed double the intended time for primary completion, which was calculated

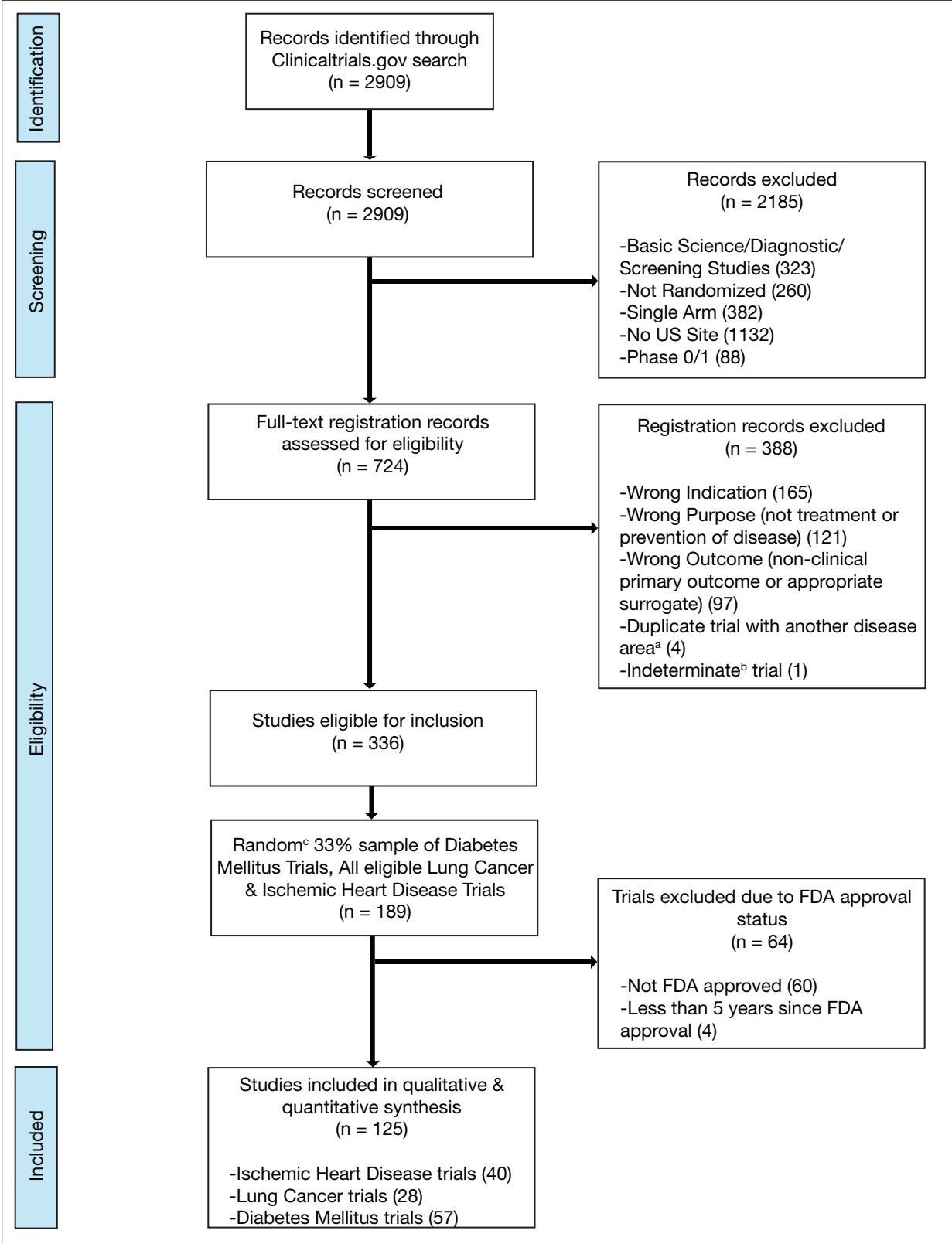

**Figure 3.** Flow diagram for trial inclusion. (**a**) Trials overlapping more than one disease area (e.g. diabetes mellitus and ischemic heart disease) were allocated based on the disease evaluated in the primary outcome. (**b**) An indeterminate trial is an ongoing trial that has not surpassed twice the planned primary outcome completion date. (**c**) We used a random number generator (RAND function in Microsoft Excel) to create our 33% sample.

*Figure 3 continued on next page*

*Figure 3 continued*

The online version of this article includes the following figure supplement(s) for figure 3:

**Figure supplement 1.** Flow diagram for ischemic heart disease interventional trials.

**Figure supplement 2.** Flow diagram for diabetes mellitus interventional trials.

**Figure supplement 3.** Flow diagram for lung cancer interventional trials.

by subtracting the intended primary completion date (as stated in the final registration record prior to study start) from the trial start date, then multiplying by 2.

We next assessed results reporting by determining whether primary outcome results were publicly available. Interventional clinical trials of FDA-regulated drug, biologic, or device interventions are subject to trial registration and results submission requirements outlined in Section 801 of the Food and Drug Administration Amendments Act (FDAAA 801) (*U.S. Public Law, 2007a*). Other types of trials, also contained in our cohort, are encouraged to register based on policies such as the International Committee of Medical Journal Editors (ICMJE) requirement for public registration (*De Angelis et al., 2004*), but will fall outside the scope of FDAAA 801 results submission requirements. Trials were categorized as reported if they either had primary outcome results available on ClinicalTrials.gov or in a publication (*Supplementary file 13*). When more than one publication presented primary outcome results, the earliest published report was identified and advanced to the next step of assessment. ClinicalTrials.gov results reporting and publication search were updated in October 2021 for those trials previously deemed to have not met the criteria for reporting.

Importance was scored by determining whether trial results were included in a high-quality review document designed to inform medical decision-making. To credit trials with being informative even if they produced negative results, trials were first assessed for inclusion in the results of a high-quality SR, given that SR citation practices are results neutral. We assessed for trial results citation in a Cochrane SR, Agency for Healthcare Research and Quality SR, or in an SR deemed of high quality based on a modified A MeaSurement Tool to Assess systematic Reviews (AMSTAR) score (*Supplementary file 14*). Trials not cited in the results of high-quality SRs were evaluated for inclusion in a high-quality clinical practice guideline (CPG); remaining uncited trials were then assessed for inclusion in an UpTo-Date review article (*Kluwer Logo, 2022*; *Supplementary file 15*). Trials cited in high-quality review documents were deemed to have fulfilled the importance condition. Assessment of importance was updated in October 2021 for all trials previously deemed to have not met the criterion for importance.

Finally, design was assessed by determining whether studies were at elevated ROB, using a modified Cochrane ROB tool (*Higgins et al., 2011*; *Supplementary file 16*). When available, ROB scores were extracted directly from the most recent high-quality SR identified during the assessment of trial importance. When unavailable, ROB scores were independently performed by two authors (NH and HM), with disagreements resolved by a third reviewer (JK). Information from both the primary study publication and the ClinicalTrials.gov registration record was used in our ROB assessments. Trials were deemed to have fulfilled the design condition of informativeness if all ROB elements were deemed to be of low ROB, or a majority were of low ROB with a minority of elements deemed to be of unclear ROB.

## Statistical analysis

Our primary outcome was the proportion of trials that met all four conditions of trial informativeness. We provided a 95% binomial CI for our primary outcome. We performed a sensitivity analysis on our primary outcome excluding small, pilot-type studies that would not have been designed to inform clinical decision-making. These were identified based on an anticipated participant enrollment below the lowest quartile of target enrollment for our cohort of trials. Due to concern that Phase 2 trials are less likely to inform clinical practice than trials of a higher phase, we performed a second sensitivity analysis on our primary outcome excluding Phase 1/2 and phase 2 trials.

As secondary outcomes, we estimated the proportion of trial participants who were enrolled in informative trials. We also report the proportion of informative trials in each of our three disease areas and the proportion of trials advancing across each condition of informativeness. We provided 95% binomial CI for the latter two secondary outcomes.

We compared the proportion of informative trials between disease categories and by trial sponsor using the Chi-square test (chisq.test function in R) and provided binomial CI for each stratum. We used the fisher.test function in R to perform a two-sided Fisher's exact test assessing the proportion of informative trials by type of intervention and trial phase and provided exact CI for each. We calculated inter-rater agreement rates using Cohen's kappa (*Supplementary file 17*). We defined $p<0.05$ as statistically significant. All analyses were performed using R version 4.0.2. (*R Development Core Team, 2013*).

Our study was not subject to Institutional Review Board approval, as it relied on publicly accessible data. The study protocol was prospectively registered on Open Science Framework (*Hutchinson et al., 2020*) deviations and amendments to the study protocol are detailed in *Supplementary file 18*. The code (*Hutchinson N, 2022*) and data set *Hutchinson et al., 2020* used in this analysis are available online. This study follows the Strengthening the Reporting of Observational Studies in Epidemiology (STROBE) reporting guidelines for cohort studies (*Supplementary file 19*).

# Additional information

## Competing interests

Deborah A Zarin: received payment as consultant for National Library of Medicine, NIH, for scientific advice to ClinicalTrials.gov and received grants from the Greenwall Foundation. Jonathan Kimmelman: received consulting fees from Amylyx Inc and payments from Biomarin. JK participated on Data Safety Monitoring Boards for NIAID and Ultragenyx. The other authors declare that no competing interests exist.

## Funding

| Funder | Grant reference number | Author |
|---|---|---|
| Fonds de Recherche du Québec - Santé | | Nora Hutchinson |
| Canadian Institute of Health Research | | Hannah Moyer |

The funders had no role in study design, data collection and interpretation, or the decision to submit the work for publication.

## Author contributions

Nora Hutchinson, Conceptualization, Data curation, Software, Formal analysis, Funding acquisition, Investigation, Visualization, Methodology, Writing - original draft, Writing – review and editing; Hannah Moyer, Data curation, Investigation, Writing – review and editing; Deborah A Zarin, Conceptualization, Methodology, Writing – review and editing; Jonathan Kimmelman, Conceptualization, Resources, Supervision, Funding acquisition, Methodology, Writing – review and editing

## Author ORCIDs

Nora Hutchinson http://orcid.org/0000-0003-1349-8592
Hannah Moyer http://orcid.org/0000-0002-5617-7927
Jonathan Kimmelman http://orcid.org/0000-0003-1614-6779

## Decision letter and Author response

Decision letter https://doi.org/10.7554/eLife.79491.sa1
Author response https://doi.org/10.7554/eLife.79491.sa2

# Additional files

## Supplementary files

- Supplementary file 1. Index.
- Supplementary file 2. Proportion of trials meeting each criterion for informativeness.
- Supplementary file 3. Trials not fulfilling feasibility condition. IHD: ischemic heart disease; DM:

diabetes mellitus; lung CA: lung cancer; PCD: primary completion date.

• Supplementary file 4. Trials not reported. IHD: ischemic heart disease; DM: diabetes mellitus; lung CA: lung cancer.

• Supplementary file 5. Trials not cited in clinical review documents. IHD: ischemic heart disease; DM: diabetes mellitus; lung CA: lung cancer.

• Supplementary file 6. Trials with concerns regarding design. IHD: ischemic heart disease; DM: diabetes mellitus; lung CA: lung cancer.

• Supplementary file 7. Phase 4 trials not meeting all four informativeness criteria. IHD: ischemic heart disease; DM: diabetes mellitus.

• Supplementary file 8. ClinicalTrials.gov search criteria.

• Supplementary file 9. Trial inclusion and exclusion criteria.

• Supplementary file 10. Assessment of regulatory approval status.

• Supplementary file 11. Addressing four conditions for informative clinical trials. [1]Column 'Conditions for informativeness' extracted from column 1 in eTable 1 *Zarin et al., 2019*

• Supplementary file 12. Classification of reason for termination.

• Supplementary file 13. Methodology for publication search.

• Supplementary file 14. Systematic review citation search strategy and quality assessment.

• Supplementary file 15. Clinical practice guideline and point-of-care medical database search strategies and quality assessment.

• Supplementary file 16. Operationalization of modified Cochrane risk of bias score.

• Supplementary file 17. Inter-rater agreement rates.

• Supplementary file 18. Deviations to the study protocol.

• Supplementary file 19. STROBE checklist for cohort Studies.

• MDAR checklist

## Data availability

The data set is available online on Open Science Framework (https://doi.org/10.17605/OSF.IO/3EGKU).

The following dataset was generated:

| Author(s) | Year | Dataset title | Dataset URL | Database and Identifier |
|---|---|---|---|---|
| Hutchinson N, Kimmelman J, Zarin D, Moyer H | 2022 | What fraction of trials inform clinical practice? A longitudinal cohort study of trials registered on clinicaltrials.gov | https://doi.org/10.17605/OSF.IO/3EGKU | Open Science Framework, 10.17605/OSF.IO/3EGKU |

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
