## [Editor Report]

This article constructs a rigorous four-step assessment of the informativeness of a clinical trial that measures its feasibility, reporting, importance, and risk of bias. This work is highly relevant for the class of trials for which it is defined, namely clinically directed randomized controlled trials. It could also be translated and validated in other areas, using data from a wider set of sources beyond the trial registry clinicaltrials.gov. The extended longitudinal nature of the assessment and the potential for some subjectivity limit this tool's utility to being a retrospective 'thermometer' for measuring informativeness rather than as a prospective diagnostic and/or fix for at-risk designs.

---

## [Decision Letter]

**Decision letter after peer review:**

Thank you for submitting your article "The Proportion of Randomized Controlled Trials That Inform Clinical Practice: A Longitudinal Cohort Study of Trials Registered on ClinicalTrials.gov" for consideration by *eLife*. Your article has been reviewed by 3 peer reviewers, including Philip Boonstra as Reviewing Editor and Reviewer #1, and the evaluation has been overseen by Mone Zaidi as the Senior Editor. The following individuals involved in review of your submission have agreed to reveal their identity: Jared Foster (Reviewer #2); Ioana Cristea (Reviewer #3).

Essential revisions:

All of the reviewers were positive about this work, but, unfortunately, each offered some non-overlapping critiques and suggestions. Here is my attempt at synthesizing some of the key themes from the reviews:

1) Could this methodology extend in principle beyond clinically directed randomized controlled trials and, if so, what are the main barriers to that?

2) Reviewer 3 offers a good discussion on the dependence of publicly available information. I wonder if the authors might engage with some of that reviewer's feedback regarding this, e.g. "In that sense, maybe it would be interesting for the authors to comment on how their methodology would be improved by having access to clinical trial protocols and statistical analysis plans."

3) Two reviewers had different comments related to the potential subjectivity in assessing the risk of bias. The authors may consider engaging with those comments.

4) Is this tool only conceivable as a retrospective diagnosis of informativeness or are there ways to envision this being deployed as a prospective tool to improve protocol design? If so, what are the barriers to such a deployment?

5) The comments under 'Recommendations for Authors' would seem to be straightforwardly addressed.

Below are the full reviews from all three reviewers.

*Reviewer #1 (Recommendations for the authors):*

1. Page 8, lines 197 onward, sentence starting "Terminated trials…". I presume that what I'm about to ask is the case, but can the authors confirm that if a trial did not achieve at least 85% enrollment due to stopping for toxicity or futility, it was nonetheless considered 'feasible'? I believe that it should be considered feasible in such a case. The language currently in this paragraph is unclear about this scenario.

2. How do the N=35 trials without timely/complete recruitment in Figure 2 relate to the 26 trials in Table 1 that had a non-Completed trial status?

3. Regarding the Reporting characteristic, many (or all) trials are required to register at clinicaltrials.gov, which requires the reporting of the trial's primary endpoint. Given this, what are the circumstances under which a trial didn't / couldn't report to clinicaltrials.gov?

4. Figure 4 isn't particularly efficient from an information-divided-by-whitespace criterion. These numbers could be presented tabularly in less space.

*Reviewer #2 (Recommendations for the authors):*

I have the following recommendations/questions:

1. It would be interesting to see a more extensive/detailed discussion/summary of specific reasons why trials failed to inform practice (beyond what is given in eTable 4).

2. Related to the previous comment, it would be interesting to see a detailed discussion/summary of specific reasons why designs ended up having elevated risk of bias scores (for cases where the necessary information is available to the authors).

3. It appears that only randomized trials included, but what about trials that are randomized but not comparative (most likely phase II), i.e., those where patients are randomized to one or more interventions, but there is no formal statistical comparison between arms? It seems like this type of design would not really be "aimed at informing clinical practice," and thus should not be included. Were these trials excluded from this analysis?

4. It would be helpful to see some discussion of what kinds of trials/designs were classified as being phase "NA."

5. While there was no significant difference in the proportions of informative trials across the three disease areas, the observed proportion of informative trials in lung cancer was noticeably lower than that in the other disease areas. Do the authors have a sense for why this might be the case?

6. It would be helpful to see some discussion of why 0.05 was selected as "significant."

---

## [Author Response]

Essential revisions:All of the reviewers were positive about this work, but, unfortunately, each offered some non-overlapping critiques and suggestions. Here is my attempt at synthesizing some of the key themes from the reviews:1) Could this methodology extend in principle beyond clinically directed randomized controlled trials and, if so, what are the main barriers to that?

Thank you for this comment. Our surrogate measures for each condition of trial informativeness are specific to clinically directed randomized controlled trials. For example, we evaluated importance of the clinical question based on citation of trial results in a clinical synthesizing document. This measure would not be applicable to other types of trials.

Creating alternative surrogate measures for other types of trials (for example, trials of unapproved products that are designed to inform a regulatory decision) represents an area of interest for our group and an avenue for future research but is beyond the scope of the current project, but

We added the following statement:

“Second, we applied strict inclusion/exclusion criteria when identifying our cohort of ‘clinically directed randomized controlled trials,’ thus limiting generalizability to other types of trials, including those involving diagnostics, early phase trials or interventions that do not advance to FDA approval. The latter would require different criteria, given their primary goal of informing regulatory or research decision-making. Developing surrogate measures of informativeness for other types of trials represents an avenue for future research.”

2) Reviewer 3 offers a good discussion on the dependence of publicly available information. I wonder if the authors might engage with some of that reviewer's feedback regarding this, e.g. "In that sense, maybe it would be interesting for the authors to comment on how their methodology would be improved by having access to clinical trial protocols and statistical analysis plans."

Thank you for this suggestion. We agree that evaluating study protocols and statistical analysis plans could be useful in improving our assessment of informativeness, particularly with respect to assessing trial design (providing more sources of data in the evaluation of the modified Cochrane risk of bias score), but also raising the possibility of assessing the fifth condition of trial informativeness, analytical integrity. However, we would be unable to obtain protocols and SAPs for our full trial sample, since they are not required to be uploaded to ClinicalTrials.gov until results are reported.

We added the following:

“Fourth, our evaluation is limited by the accuracy of information contained in the ClinicalTrials.gov registration record and in the published literature. The use of other sources of data, such as U.S. Food and Drug Administration regulatory documents, in addition to study protocols and statistical analysis plans uploaded onto ClinicalTrials.gov, could be considered for use in future evaluations of trial informativeness.”

3) Two reviewers had different comments related to the potential subjectivity in assessing the risk of bias. The authors may consider engaging with those comments.

We extracted risk of bias scores from the systematic review that cited primary outcome results of the index trial. This applied to 36 trials (57% of trials assessed for design). We did not assess whether risk of bias assessments in one review differed from another, which is a limitation. When no risk of bias assessment was performed, two authors independently performed risk of bias assessments using information from both the primary study publication and the ClinicalTrials.gov registration record. Our unweighted Cohen’s kappa for assessment of design was 0.84 (Supplementary File 17).

We have added the following sentence to the Materials and methods section to clarify the sources of our risk of bias assessment:

“When available, ROB scores were extracted directly from the most recent high-quality SR identified during the assessment of trial importance. When unavailable, ROB scores were independently performed by two authors (NH and HM), with disagreements resolved by a third reviewer (JK). Information from both the primary study publication and the ClinicalTrials.gov registration record were used in our risk of bias assessments.”

We have also added a table in the supplement (Supplementary File 6) which provides the risk of bias assessments for the 30 trials that did not meet our design condition.

4) Is this tool only conceivable as a retrospective diagnosis of informativeness or are there ways to envision this being deployed as a prospective tool to improve protocol design? If so, what are the barriers to such a deployment?

This is a question our team has discussed throughout this project. Our conditions for informativeness, as currently conceived, enable retrospective trial assessment. For example, outcomes like inclusion in a clinical practice guideline or systematic review are only assessable post facto. However, the information gathered here could nonetheless be used to prospectively identify certain types of trials that would benefit from greater oversight to improve trial informativeness (e.g., our data seem to flag non-industry funded or phase IV trials as being more at risk of being uninformative).

The ultimate goal, however, would be to create a tool that can be used prospectively to improve trial informativeness. Our current model is a first step towards achieving this goal – if we can assess a cohort of trials using retrospective conditions for informativeness, and achieve agreement in the scientific community that our measures are meaningful, we can use this information to build a prospective model. For example, protocols might present evidence of recruitment feasibility or a reporting plan that IRBs or others might use to assess prospects of fulfilling informativeness.

We changed the following sentence in the Introduction to clarify that our current model involves a retrospective evaluation of trial informativeness:

“Our approach involved a retrospective evaluation of trial informativeness. Findings can help healthcare and research systems identify study types in need of further scrutiny, thereby improving the impact of future research.”

We also added the following sentences to the end of the Discussion:

“Although there is broad agreement that uninformative trials exist, there is no clear consensus on methods to identify which trials are uninformative. Our retrospective assessment is an initial step towards developing prospective criteria that can be used to highlight trials of concern, thereby enabling early corrective intervention.”

5) The comments under 'Recommendations for Authors' would seem to be straightforwardly addressed.

Thank you. Each is addressed below.

Below are the full reviews from all three reviewers.Reviewer #1 (Recommendations for the authors):1. Page 8, lines 197 onward, sentence starting "Terminated trials…". I presume that what I'm about to ask is the case, but can the authors confirm that if a trial did not achieve at least 85% enrollment due to stopping for toxicity or futility, it was nonetheless considered 'feasible'? I believe that it should be considered feasible in such a case. The language currently in this paragraph is unclear about this scenario.

Yes, that is correct. We have clarified the description of our feasibility assessment in the Materials and methods section:

“Terminated trials were deemed infeasible if the reason for termination in the ClinicalTrials.gov registration record involved accrual, feasibility, funding or another non-scientific reason (Supplementary File 12). Trials terminated for scientific reasons (accumulated scientific data suggesting early efficacy, futility or toxicity) were deemed feasible irrespective of the proportion of expected enrollment achieved.”

2. How do the N=35 trials without timely/complete recruitment in Figure 2 relate to the 26 trials in Table 1 that had a non-Completed trial status?

We address this in the Supplement by adding Supplementary File 3 – Trials Not Fulfilling Feasibility Condition. It shows that, of the 35 trials not fulfilling the feasibility condition, 30 had reached a final outcome of “Completed” or “Terminated”, 3 were of status “Unknown” and 2 were “Active, Not Recruiting.” We are happy to bring this into the main text if the referee or editors think it would clarify messaging.

3. Regarding the Reporting characteristic, many (or all) trials are required to register at clinicaltrials.gov, which requires the reporting of the trial's primary endpoint. Given this, what are the circumstances under which a trial didn't / couldn't report to clinicaltrials.gov?

We have added Supplementary File 4 – Trials Not Reported to the supplement and provide reasons for why a trial may not have been reported. Trials that are subject to FDAAA 801 are required to both register and report results through ClinicalTrials.gov. However, many trials that fall outside the scope of FDAAA 801 would be incentivized to register (based on other policies; for example, the International Committee of Medical Journal Editors (ICMJE) requirement for public registration) but may not be required to make results publicly available.

We added the following sentence to the Materials and methods section:

“Interventional clinical trials of Food and Drug Administration-regulated drug, biologic or device interventions are subject to trial registration and results submission requirements outlined in Section 801 of the Food and Drug Administration Amendments Act (FDAAA 801). Other types of trials, also contained in our cohort, are encouraged to register based on policies such as the International Committee of Medical Journal Editors (ICMJE) requirement for public registration, but will fall outside the scope of FDAAA 801 results submission requirements.”

4. Figure 4 isn't particularly efficient from an information-divided-by-whitespace criterion. These numbers could be presented tabularly in less space.

Thank you for this suggestion. We have replaced Figure 4 with Table 2.

Reviewer #2 (Recommendations for the authors):I have the following recommendations/questions:1. It would be interesting to see a more extensive/detailed discussion/summary of specific reasons why trials failed to inform practice (beyond what is given in eTable 4).

Thank you for this suggestion! We have added a table in the supplement for each of the conditions of informativeness assessed and provide information about every trial not fulfilling each condition (Supplementary File 3 – Trials Not Fulfilling Feasibility Condition; Supplementary File 4 – Trials Not Reported; Supplementary File 5 – Trials Not Cited in Clinical Review Documents; and Supplementary File 6 – Trials with Concerns Regarding Design).

We have also added the following to the Results section:

“Of the 35 trials not meeting our feasibility condition, 21 failed to reach 85% of planned patient enrollment (Supplementary File 3). Of the 9 trials not meeting our reporting condition, 6 were not subject to FDAAA 801 results reporting requirements (Supplementary File 4). Although we cannot be certain why a trial was not incorporated into a clinical synthesizing document, possible reasons are illustrated by the following examples. For one trial (NCT01104155 – phase 2 study in non-small cell lung cancer investigating the combination of eribulin mesylate in combination with intermittent erlotinib), increased understanding of the importance of biomarker status in treatment response from trial initiation to completion rendered the results for a non-selected population less clinically relevant (Supplementary File 5). In another case, trial NCT00918138 investigated the addition of saxagliptin to extended-release metformin in adult patients with type 2 diabetes. Its primary outcome, change from baseline in 24 hour mean weighted glucose at week 4, did not meet eligibility criteria for inclusion in a high-quality systematic review investigating the same topic. In a third case, trial NCT00954707 (a phase 4 trial investigating duration of dual anti-platelet therapy in 2509 individuals undergoing placement of a Cypher cardiac drug eluting stent) sponsors submitted results to ClinicalTrials.gov but never published them. This may have hampered inclusion in a clinical synthesizing document. Eight of the 18 trials not meeting the importance condition had no published primary outcome results. Finally, the most common reasons for a high risk of bias score in the 30 trials not meeting our design condition were lack of blinding of participants and personnel, lack of blinding of outcome assessment and selective reporting (Supplementary File 6).”

2. Related to the previous comment, it would be interesting to see a detailed discussion/summary of specific reasons why designs ended up having elevated risk of bias scores (for cases where the necessary information is available to the authors).

Please see Supplementary File 6 – Trials with Concerns Regarding Design, and our response to question 2.

3. It appears that only randomized trials included, but what about trials that are randomized but not comparative (most likely phase II), i.e., those where patients are randomized to one or more interventions, but there is no formal statistical comparison between arms? It seems like this type of design would not really be "aimed at informing clinical practice," and thus should not be included. Were these trials excluded from this analysis?

We included clinical trials that were indicated to be multi-arm on ClinicalTrials.gov. Although these might theoretically include basket or other master protocol designs, we did not have any in our sample. We did have two studies in our cohort (NCT01183858 and NCT01104155) that each had two study arms comparing efficacy of different doses of a drug, without additional comparator arms. Of these two trials, one (NCT01183858) met all of our 4 conditions for an informative trial.

4. It would be helpful to see some discussion of what kinds of trials/designs were classified as being phase "NA."

Technically, “phase” only applies to studies that involve drugs/biologics. The following definition of “Phase Not Applicable” is provided in the glossary of ClinicalTrials.gov (https://clinicaltrials.gov/ct2/about-studies/glossary): “Describes trials without FDA-defined phases, including trials of devices or behavioral interventions.”

Our cohort of trials included 32 trials of phase “NA.” Of these trials, 3 involved drug interventions that were likely mislabeled as phase NA. The remainder included the following types of interventions: behavioral interventions (education, case management, rehabilitation, communication aids, intervention protocols), procedures/surgical interventions and devices.

We have added the following to the legend of Table 1 with the intention of adding clarity to the phase NA designation: (line 118-119)

“Includes behavioral, procedural/surgical or device interventions .”

5. While there was no significant difference in the proportions of informative trials across the three disease areas, the observed proportion of informative trials in lung cancer was noticeably lower than that in the other disease areas. Do the authors have a sense for why this might be the case?

Thank you for noticing this. There was a notable reduction in informative lung cancer trials after assessment of trial design. Many were noted to be at high risk of bias due to lack of blinding of patients and participants and/or blinding of outcome assessment. There was also a higher proportion of phase 2 lung cancer trials (46.4%) in comparison to ischemic heart disease (15.0%) and diabetes mellitus trials (8.8%). These differences in lung cancer trial design contributed to their lower overall proportion of informative trials (14.3%) seen in this group.

We have added the following sentence to our Discussion:

“Although there was no significant difference in informativeness observed between our three disease areas, many lung cancer trials were of early phase (46.4% phase 2) and did not meet design criteria due to lack of blinding (Supplementary File 6). We acknowledge that for some of these trials, blinding may have been difficult to achieve but nonetheless would contribute to an elevated risk of bias. Of note, trials were assessed as “low risk of bias” if there was no or incomplete blinding, but the outcome was unlikely to be influenced by lack of blinding, as per the Cochrane ROB guidelines.”

6. It would be helpful to see some discussion of why 0.05 was selected as "significant."

This was based on convention, as proposed by the statistician RA Fisher (Fisher, R.A. The arrangement of field experiments. Journal of the Ministry of Agriculture. 1926;33: 503-515.) To us it seemed reasonable for a ‘first of its type’ study of this sort.